# Comparison of diagnostic algorithms used in guidelines on nutritional anaemias in adults

Margaretha G. M. Roemer[1,2☯], Wendy P. J. den Elzen[3,4☯], Rosalinde K. E. Poortvliet[5,6], Jacobijn Gussekloo[5,6,7], Steef Kurstjens[8,9], Bauke A. de Boer[1], Anne Margreet de Jong [1]*

1 Department of Clinical Chemistry, Unilabs/ Atalmedial Medical Diagnostic Centers, Amsterdam, the Netherlands, 2 Department of Clinical Chemistry, Haematology & Immunology, Noordwest Ziekenhuisgroep, Alkmaar, the Netherlands, 3 Laboratory Specialized Diagnostics & Research, Department of Laboratory Medicine, Amsterdam UMC, University of Amsterdam, Amsterdam, the Netherlands, 4 Amsterdam Public Health Research Institute & Amsterdam Gastroenterology and Endocrinology Metabolism, Amsterdam, the Netherlands, 5 Department of Public Health and Primary Care, Leiden University Medical Center, Leiden, the Netherlands, 6 LUMC Center for Medicine for Older people, Leiden, the Netherlands, 7 Section of Gerontology and Geriatrics, Department of Internal Medicine, Leiden University Medical Center, Leiden, the Netherlands, 8 Laboratory of Clinical Chemistry and Hematology, Jeroen Bosch Hospital, 's Hertogenbosch, the Netherlands, 9 Laboratory of Clinical Chemistry and Laboratory Medicine, Dicoon BV, location Canisius Wilhelmina Hospital, Nijmegen, the Netherlands

☯ These authors contributed equally to this work.
* anne-margreet.dejong@unilabs.com

## Abstract

### Background

Anaemia is highly prevalent and commonly caused by nutritional deficiencies. Guidelines often include an algorithm to find the underlying cause. Here, we compared diagnostic algorithms and suggested laboratory tests for anaemia in clinical practice guidelines in countries with similar healthcare systems, focusing on iron, vitamin B12 and folate deficiency in the general adult population.

### Methods

We searched for diagnostic guidelines on anaemia in the Trip, Guidelines International Network and country specific databases. To be selected, the guidelines had to include diagnostic criteria or algorithms to determine the cause of anaemia in the general adult population.

### Results

In total, 14 records were included. For iron deficiency anaemia, guidelines varied in diagnostic criteria ranging from use of ferritin only, to ferritin in various combinations with a variety of other parameters, with different cut-off values. For vitamin B12 or folate deficiency, besides measurement of vitamin B12 or folate, some guidelines mentioned methylmalonic acid or homocysteine. Quality of evidence

**Data availability statement:** All relevant data are within the paper and its Supporting information files.

**Funding:** The author(s) received no specific funding for this work.

**Competing interests:** The authors have declared that no competing interests exist.

underlying cut-offs and parameters was variable, and laboratory aspects were underrepresented.

## Conclusions

There was a lot of variation in the included diagnostic algorithms, especially for iron deficiency anaemia. Differences in cut-off values were seen, even when using similar diagnostic strategies. Furthermore, supporting evidence was variable. Further research is needed to determine optimal algorithms. Our findings highlight the need for inclusion of relevant laboratory aspects in guidelines, appropriate diagnostics and clinical decision limits.

---

## Introduction

Anaemia is a highly prevalent disorder worldwide affecting up to 25% of the population [1,2], and is defined as a low number of red blood cells or a low haemoglobin concentration within the red blood cells for the age and sex of the individual [1]. Low haemoglobin concentrations result in a decreased capacity of the blood to carry oxygen to the tissues in the body. Consequently, symptoms include fatigue, weakness, dizziness and shortness of breath. Anaemia is not a diagnosis in itself, but often a presentation of an underlying condition. Anaemia is most frequently caused by nutrient deficiencies due to inadequate diets or inadequate absorption of nutrients, predominantly iron, vitamin B12 and folate. In addition, chronic diseases, infection, kidney failure, genetic disorders, bone marrow pathologies, and acute or chronic haemorrhage can cause anaemia [3,4]. It is therefore relevant to correctly diagnose the underlying cause of anaemia to adequately treat the patient.

General practitioners play a crucial role in the diagnosis and management of anaemia. The diagnosis typically involves a complete blood count (CBC) and further investigative assays to pinpoint the specific aetiology. Premature or unnecessary testing can lead to excessive use of diagnostic blood tests, while failure to test anaemic patients can result in incomplete or missed diagnoses or under- or overtreatment. Both scenarios can potentially increase healthcare costs [5–7].

To ensure effective and standardised care, anaemia guidelines have been developed. These guidelines typically encompass initial assessment, diagnostic testing, management and treatment, follow-up and monitoring. For different physiological settings, including age, sex, and pregnancy status or for specific patient groups, different guidelines for diagnosing anaemia have been developed. Also, different countries have varying requirements for the way in which guidelines are developed, who is involved in their development, and what sources are used. Therefore, the aim of this review was to compare the diagnostic algorithms and suggested laboratory tests for anaemia in clinical practice guidelines. Countries with similar healthcare systems were selected and we focused on the most common causes of anaemia: iron, vitamin B12 and folate deficiency in adults from the primary care population. This may provide guidance for the development or update of (novel) guidelines.

## Methods

### Data sources and searches

First, we searched for guidelines on anaemia in two guideline databases: Trip database (https://www.tripdatabase.com) and Guidelines International Network (G-I-N, http://www.g-i-n.net) on January 12th 2022. These databases were selected to search for evidence-based guidelines across the world. We performed an advanced search for 'anemia' or 'anaemia' in both databases. Second, we performed searches in country specific guideline databases (ao. Germany, UK, USA, Canada, Australia) on October 15th 2023. The most recent version of the guidelines was sought and used on February 29th 2024.

### Guideline selection

To be selected, the guidelines had to include diagnostic criteria or algorithms to find the cause of anaemia in the primary care adult population. We focused on nutritional deficiencies (iron, vitamin B12 and folate). Guidelines had to include criteria for adults and specific guidelines for children were excluded. When multiple guidelines from the same society were found, the most recent version was used. Guidelines focusing on specific populations (e.g., anaemia in pregnancy, chronic kidney disease, dialysis patients, heart failure, cancer), guidelines of which full versions were not available in English, Dutch or German and guidelines of which full versions were inaccessible online or withdrawn, were excluded. Two authors (combinations of MR, WdE, AdJ) independently assessed the titles and full texts of the guidelines retrieved during the searches against the inclusion and exclusion criteria, and selected or rejected as appropriate. Disagreement was resolved by consensus.

### Data extraction, quality assessment and analysis

The recommendations in the selected guidelines were extracted by two authors (WdE, AdJ) and checked by a third (MR). We rated the quality of the guidelines with a scoring system, based on the evidence underpinning the recommendations. Level 0: no evidence provided, level 1: based on expert opinion, level 2: based on randomised controlled trials/ peer reviewed studies before 2010 (for the test of interest), level 3: based on randomised controlled trials/ peer reviewed studies 2010 and later (for the test of interest), level 4: based on a systematic review/ meta-analysis. Note that guidelines that referred to studies before, as well as after 2010 were evaluated as level 3.

## Results

A total of 87 records were identified by the combined searches (S1 Table). Forty-four records were excluded due to the focus on a specific patient group. The remaining 43 full texts were screened and another 29 were excluded in line with the exclusion criteria (Fig 1). The 14 included guidelines originated from 6 different countries (US, UK, Germany, Canada, Qatar and the Netherlands).

### Patient population

We focused on guidelines in the primary care patient population. Nonetheless, a plethora of different anaemia guidelines were found that included specific organs, patient groups and underlying diseases. Therefore, we decided to select all guidelines focusing on the "general population". This included primary care, as well as secondary care guidelines in otherwise healthy people.

Age definitions for patient populations that were included in the different guidelines were variable. Most of the guidelines did not state a specific age definition [8–15], while others focused on age ≥ 5 years [16], > 12 years [17], ≥ 16 years [18,19], ≥ 18 years [20] or on older adults [21]. Although all guidelines acknowledged that older people have a higher risk of developing anaemia, and/or suggested further diagnostics, no age-specific diagnostic algorithms were provided.

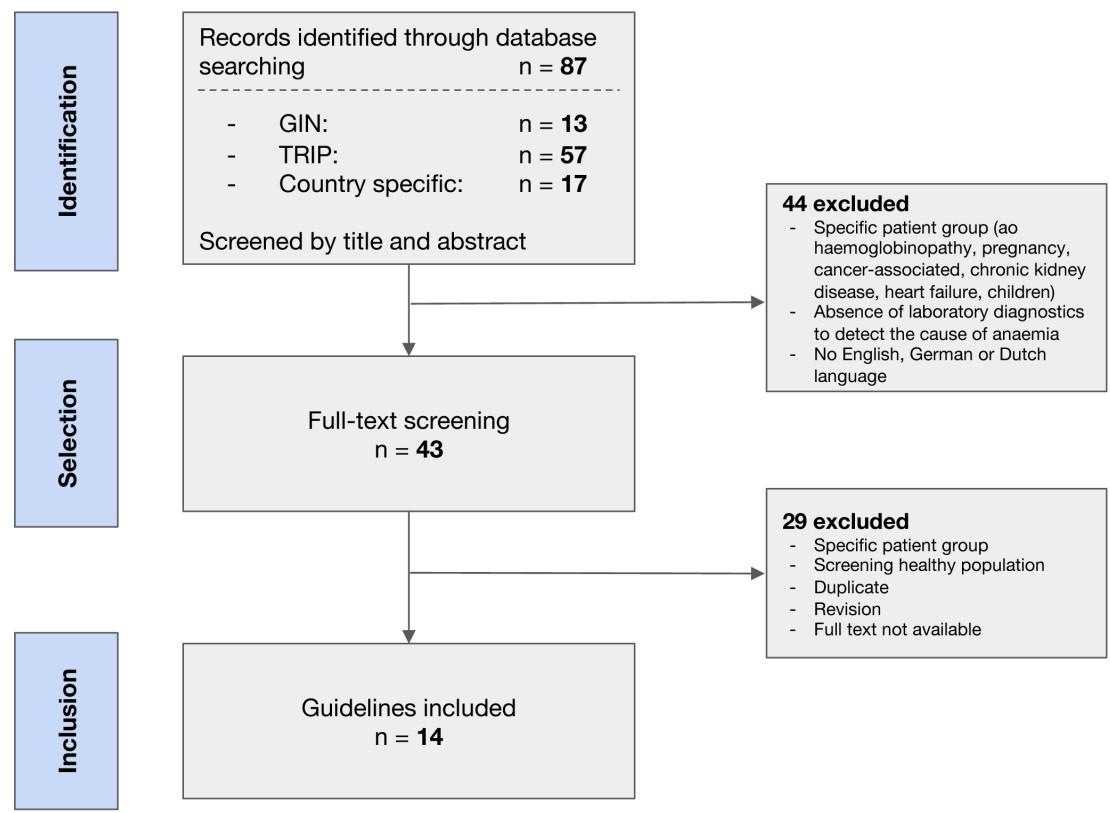

**Fig 1. Flowchart of selected guidelines.**

## Definition of anaemia

According to the World Health Organization (WHO), anaemia is defined as a haemoglobin level <12.0 g/dL in women and <13.0 g/dL in men [1]. Most of the guidelines adhered to this definition [8,9,11–14,17,19,20], although one guideline used <13.5 g/dL in men [16] and one guideline used <11.5 g/dL in women (Table 1) [18]. Three guidelines did not specify a value for low haemoglobin, but referred to local laboratory reference values [10,15,21].

## Iron deficiency anaemia

For iron deficiency anaemia, there was a lot of variation in the included diagnostic criteria, as summarised in Table 2. Diagnostic criteria ranged from only ferritin, to ferritin in various combinations with mean corpuscular volume (MCV), iron, transferrin, transferrin saturation, total iron binding capacity (TIBC), C-reactive protein (CRP),erythrocyte sedimentation rate (ESR), reticulocyte haemoglobin content (Ret-Hb) and reticulocyte count, with different cut-off values. Of note, even when a similar strategy was used, differences in cut-off values were seen.

Testing of MCV was recommended in seven guidelines, in combination with ferritin [9–12,16,20,21]. Other erythrocyte indices (mean corpuscular haemoglobin, mean corpuscular haemoglobin concentration, red cell distribution width) were variably mentioned in a minority of the guidelines, and therefore not included in Table 2.

All guidelines included ferritin for the diagnosis of iron deficiency anaemia. Cut-offs ranged from <15 µg/L [15,18] to <30 µg/L [8,14,17,20] and <45 µg/L [13] or referred to the local lower limit of reference range [10–12,21]. Two guidelines used different ferritin cut-offs for men and women [9,16].

**Table 1. Summarized haemoglobin cut-offs in the included guidelines to define anaemia.**

| Guideline | Year | Country | Haemoglobin level | Evidence | Ref |
|---|---|---|---|---|---|
| NHS Greater Glasgow and Clyde | 2022 | UK | M < 13 g/dL, F < 11.5 g/dL | 1 | 18 |
| Centre for Perioperative Care | 2022 | UK | M < 13 g/dL, F < 12 g/dL * | 4 | 8 |
| Association of Scientific Medical Societies Germany | 2021 | Germany | M < 13 g/dL, F < 12 g/dL | 4 | 9 |
| British Society of Gastroenterology | 2021 | UK | < N | NA | 10 |
| National Institute for Health and Care Excellence | 2021 | UK | M < 13 g/dL, F < 12 g/dL | 4 | 17 |
| Dutch College of General Practitioners | 2021 | The Netherlands | M < 13 g/dL, F < 12 g/dL # | 4 | 11 |
| Ministry of Public Health Qatar | 2020 | Qatar | M < 13 g/dL, F < 12 g/dL | 4 | 12 |
| American Gastroenterological Association | 2020 | USA | M < 13 g/dL, F < 12 g/dL | 4 | 13 |
| National Institute for Health and Care Excellence | 2020 | UK | M < 13 g/dL, F < 12 g/dL # | 4 | 19 |
| Royal College of Nursing | 2019 | UK | M < 13 g/dL, F < 12 g/dL | 4 | 14 |
| BC guidelines | 2019 | Canada | < N | NA | 15 |
| Association of Scientific Medical Societies Germany | 2018 | Germany | M < 13 g/dL, F < 12 g/dL, Drop > 2 g/dL | 4 | 20 |
| Toward Optimized Practice | 2018 | Canada | M < 13.5 g/dL, F < 12 g/dL # | 1 | 16 |
| American Medical Directors Association | 2007 | USA | < N | NA | 21 |

Abbreviations: Ref, reference; M, male; F, female; N, normal; NA, not applicable.

* international consensus statement in the peri-operative period: Hb < 13 g/dL for both sexes [22].

# Suggested cut-off values for assessment.

The fact that ferritin is an acute phase protein and thus increases with infection or inflammation was recognized in all guidelines. Nonetheless, some guidelines only mentioned it [9,12,13,17,21], while other guidelines specified the ferritin level for which iron deficiency could still be considered in the presence of inflammation [11,15,16,18,20]. Measurement of CRP [8,14]. or ESR and/or CRP [10] was recommended in three guidelines to confirm the presence of inflammation. Additional testing of transferrin saturation when a normal ferritin level was found was recommended in five guidelines [8,10,14,20,21].

Testing of the serum iron was recommended in three guidelines, in combination with MCV, ferritin, transferrin, transferrin saturation and/or TIBC [11,12,21]. Lastly, three guidelines recommended examination of Ret-Hb or reticulocyte count [8–10].

### Vitamin B12 or folate deficiency anaemia

Six vitamin B12 and folate deficiency anaemia guidelines were included. Most of these guidelines contained a combined anaemia diagnostic algorithm with iron deficiency [8,11,12,20,21] and one guideline specifically focused on vitamin B12 and folate deficiency (Table 3) [19]. All guidelines mentioned that vitamin B12 and/or folate anaemia is macrocytic. Four guidelines stated that the anaemia may also be normocytic [8,11,19,21].

For vitamin B12, one guideline recommended a specific cut-off [19], the other guidelines referred to local laboratory reference values. With regards to the use of total vitamin B12 (cobalamin) or active vitamin B12 (holotranscobalamin), four guidelines specifically recommended the (initial) testing of total vitamin B12 [8,11,19,20].

**Table 2. Summarized diagnostic algorithms of the included iron deficiency anaemia guidelines.**

| Guideline | Year | Country | MCV (fl) | Ferritin (µg/L) | Iron | Trans-ferrin | Transferrin sat (%) | TIBC | CRP (mg/L) or ESR | Ret-He/ Reticulocyte count | Evi-dence | Ref |
|---|---|---|---|---|---|---|---|---|---|---|---|---|
| NHS Greater Glasgow and Clyde | 2022 | UK | | < 15 or 15–50 † | | | | | | | 1 | 18 |
| Centre for Perioperative Care | 2022 | UK | | < 30 * or 30–100 † | | | <20% * | | > 5 † | < 30 pg * | 3 | 8 |
| Association of Scientific Medical Societies Germany | 2021 | Germany | < 78 | M<35, F<23 or ≥ N † | | | | | | < 28 pg/ ≤ N | 2 | 9 |
| British Society of Gastroenterology | 2021 | UK | < N* | < N or N * | | | Low * | | > N * | < N * | 4 | 10 |
| National Institute for Health and Care Excellence | 2021 | UK | | < 30 ‡ | | | | | | | 3 | 17 |
| Dutch College of General Practitioners | 2021 | the Neth-erlands | ≤ N | < N or < 100 †* | < N * | > N * | | | | | 2 | 11 |
| Ministry of Public Health Qatar | 2020 | Qatar | < N | ≤ N ‡ | < N | | | ≥ N | | | 3 | 12 |
| American Gastroenterological Association | 2020 | USA | | < 45 ‡ | | | | | | | 4 | 13 |
| Royal College of Nursing | 2019 | UK | | < 30 or 30–100 ^ | | | < 20% ^ | | > 5 ^ | | 1 | 14 |
| BC guidelines | 2019 | Canada | | < 15 or 15–30 $ or < 100 † | | | | | | | 2 | 15 |
| Association of Scientific Medical Societies Germany | 2018 | Germany | < N | <30 * or 30 - 100 * or> 300 †# | | | < 20% * or †# | | | | 3 | 20 |
| Toward Optimized Practice | 2018 | Canada | < 75 * | M<30 *, F<13 or < 100 † | | | | | | | 3 | 16 |
| American Medical Directors Association | 2007 | USA | < N | < N or < N # or N † | < N #† | > N # | < N #† | > N # | | | 2 | 21 |

Abbreviations: MCV, mean corpuscular volume; sat, saturation; TIBC, total iron binding capacity; CRP, C-reactive protein; ESR, erythrocyte sedimentation rate; Ret-He, reticulocyte haemoglobin equivalent; Ref, reference; M, male; F, female; N, normal; pg, picograms.

* and/ or one of the other parameters with *,

# and,

^ or,

† Patients with chronic (inflammatory) disease, active inflammation or malignancy,

‡ Ferritin is an acute phase protein and increases in patients with infection or inflammation,

$ Probably iron deficiency.

In addition to vitamin B12, four guidelines mentioned testing for methylmalonic acid (MMA) and/or homocysteine, always in addition to vitamin B12 [12,21] or when the vitamin B12 level is in the lower normal reference range [11,20].

For folate, most guidelines recommended testing for serum folate only [8,11,21]. Two guidelines included RBC folate as a diagnostic criterion [19,20]. One guideline recommended measurement of lactate dehydrogenase (LDH) and reticulo-cyte count to diagnose folate, as well as vitamin B12 deficiency [11].

## Quality of the evidence

The quality of the included guidelines was evaluated and ranked for the indicated parameters (colour-coded in Tables 1–3). To define anaemia, WHO haemoglobin cut-offs were used in most guidelines [8,9,11–14,17,19–21]. Given the systematic nature of the evidence and recently unchanged cut-offs [23], this was regarded as level 4. For iron deficiency anaemia, two guidelines were considered level 4 [10,13], five guidelines level 3 [8,12,16,17,20], four guidelines level 2

**Table 3. Summarized diagnostic algorithms of the included vitamin B12/ folate deficiency anaemia guidelines.**

| Guideline | Year | Country | MCV (fl) | Vit. B12 (pmol/L) | Active vit. B12 | MMA | Hcy | Folate (nmol/L) | RBC folate (nmol/L) | MMA | Hcy | LDH | Reticulo-cyte count | Evi-dence | Ref |
|---|---|---|---|---|---|---|---|---|---|---|---|---|---|---|---|
| | | | | **Vitamin B12** | | | | **Folate** | | | | **Folate and vit. B12** | | | |
| Centre for Perioper-ative Care | 2022 | UK | ≥ N | < N | | | | < N | | | | | | 3 | 8 |
| Dutch College of General Practitioners | 2021 | the Nether-lands | ≥ N | < N or low N * | | Optional (1st) * | Optional (2nd) * | < N | | | | > N | < N | 3 | 11 |
| National Institute for Health and Care Excellence | 2020 | UK | ≥ N | < 148 | | | | < 7 or N # | < 340 # | | | | | 3 | 19 |
| Ministry of Public Health Qatar | 2020 | Qatar | > N | < N | | > N | | < N | | N | | | | 3 | 12 |
| Association of Scientific Medical Societies Germany | 2018 | Ger-many | > N | < N or low N * | Optional * | Optional * | Optional * | < N ^ | Confirm | | Optional | | | 3 | 20 |
| American Med-ical Directors Association | 2007 | USA | ≥ N | < N | | > N | > N | < N | | | | | | 2 | 21 |

Abbreviations: MCV, mean corpuscular volume; Vit, vitamin; MMA, methylmalonic acid; Hcy, homocysteine; RBC, red blood cell; LDH, lactate dehydro-genase; Ref, reference; N, normal.

* and/ or one of the other parameters with *,

# with normal vit. B12.

^ related to food intake, a second fasting blood test is required.

[9,11,15,21] and two guidelines level 1 [14,18]. It must be noted that guidelines frequently referred to each other, but none-theless used variable cut-offs. Five out of six vitamin B12 and folate deficiency anaemia guidelines were considered level 3 [8,11,12,19,20] and one guideline level 2 [21].

## Laboratory aspects

Laboratory aspects were only scarcely reported in the included guidelines. Even though laboratory measurements are used in all guidelines, none of the guidelines specified which manufacturer or measurement method was used to deter-mine the specific cut-off values, and the type of material (e.g., plasma or serum) that should be used for the test of interest.

## Discussion

In primary care, the CBC is one of the most common blood tests, often done as a routine check-up. When a low haemo-globin is found, follow-up tests are used to investigate the cause of anaemia [24]. Here, we elucidated similarities and differences in diagnostic algorithms in evidence-based guidelines for the most common causes of anaemia: iron, vitamin B12 and folate deficiency. An overview of our observations and recommendations for future guidelines is summarised in Fig 2. In brief, we observed that primary and secondary care guidelines are intertwined, and that there is insufficient atten-tion to diversity aspects. Furthermore, a lot of variation was seen in the recommended diagnostic criteria, cut-off values and evidence underpinning the algorithms, even in the presence of similar diagnostic strategies. Importantly, even though laboratory testing is proposed in all guidelines, the specifics of these tests are not described.

| Observation in current guidelines | Recommendation for future guidelines |
|---|---|
| 1. Plethora of different anaemia guidelines per organ, patient group and underlying disease for primary and secondary care | 1. Develop guidelines specifically for the primary and secondary care settings and align these |
| 2. No specific attention is paid to individual subgroups, such as age and sex | 2. Include aspects of diversity (at least age and sex) in diagnostic algorithms |
| 3. Variation in algorithms in diagnostic criteria and cut-off values | 3. Further research is needed to determine optimal diagnostic algorithms |
| 4. Most guidelines focus on inclusion of MCV | 4. MCV is not the most important diagnostic parameter, use MCV as a supportive parameter only |
| 5. Variation in used cut-off values for ferritin | 5. The test for ferritin is not harmonized: refer to local laboratory reference ranges instead of mentioning actual cut-off values |
| 6. Variable views on how to deal with inflammation in patients with iron deficiency | 6. Mention the importance of acknowledging the presence of inflammation, especially for ferritin |
| 7. Not clear if total or active vitamin B12 should be used | 7. Align the recommended test for vitamin B12 deficiency |
| 8. Lack of and/or variable quality of evidence | 8. Critically evaluate quality of evidence to include in guideline: rigorous methodology and well-defined approach |
| 9. Little to no awareness of laboratory aspects in guidelines<br>   a. Different platforms produce different results<br>   b. No harmonization / standardization of tests | 9. Specifically mention laboratory aspects in guidelines<br>   a. Specify the platform that is used to determine cut-offs<br>   b. When there is no harmonization / standardization of the test: refer to local laboratory reference ranges |

**Fig 2. Observations in current guidelines and recommendations for future guideline revisions with respect to anaemia.**

## Patient population and diversity in diagnostic algorithms

Primary care, as well as secondary care guidelines for otherwise healthy people were included. For the development or update of (novel) guidelines, a focus on either the primary or secondary care setting is endorsed, as well as alignment between the recommendations of primary care and secondary care. Similar misalignments were observed in the field of subclinical hyperthyroidism [25].

Remarkably, no distinctions were made by sex or age in most of the diagnostic algorithms. However, research shows that sex differences may be of importance [26]. Nonetheless, the WHO did not find enough evidence for a change of ferritin cut-off values by sex [27]. Additionally, the diagnostic work-up and treatment of anaemia in older patients may be different from other age groups [28–30]. Hence, we recommend including at least age and sex, in the diagnostic algorithms.

## Included parameters in diagnostic algorithms

After anaemia is identified, most of the guidelines used the MCV to define the cause of the anaemia. However, a normal MCV does not exclude the presence of iron, vitamin B12 or folate deficiency [31]. Therefore, MCV should be mainly used as a supporting parameter.

For iron deficiency anaemia, ferritin was included in all algorithms. However, the cut-offs were variable, as has been previously described [32–35] and no proper justification for these cut-offs was provided. Despite an extensive review performed by the WHO, insufficient evidence was found for a consensus on the ferritin threshold and therefore the cut-off of <15 µg/L remained unchanged [27,32,33]. In the presence of inflammation, WHO recommends a cut-off of 70 µg/L to diagnose iron deficiency anaemia [27]. However, most guidelines recommended a ferritin cut-off of 100 µg/L in this situation. There is no consensus about which additional parameters should be evaluated. Furthermore, ferritin levels are affected by recent oral iron supplementation intake, coffee and other dietary factors [36]. Therefore, it is recommended to fast overnight and withhold supplements before measuring ferritin [37]. These factors are not adequately emphasized in guidelines, which can lead to diagnostic confusion.

For vitamin B12 and folate deficiency, most guidelines referred to local lower limit of reference values. Given the variation of used cut-offs in literature [38,39], this is a sensible approach. Not all guidelines specify if total vitamin B12 or active vitamin B12 should be tested. Also, additional testing of MMA and homocysteine is not consistently advised. To date, the most reliable parameter to detect vitamin B12 deficiency is still a matter of debate. According to Herbert's model (1986), active vitamin B12 is an earlier and more sensitive indicator of vitamin B12 deficiency [40,41], which was supported by later studies [42–45]. Nonetheless, other studies contradict this and state that the reliability of active and total vitamin B12 is similar [46,47].

These findings highlight the need for additional research to determine the most optimal parameters and cut-offs to diagnose anaemia caused by nutritional deficiencies.

## Laboratory aspects in guidelines

There is limited awareness of the laboratory aspects in anaemia guidelines, although all the algorithms use laboratory tests to determine the cause of anaemia. It must be noted that the ferritin, vitamin B12 and folate assays are not harmonised between platforms and there are substantial inter-analyser differences. The lack of harmonisation may result in different values when the same sample is measured in different labs using different analyser platforms [48–52]. Concentrations may also be dependent on the type of material, e.g., serum or plasma that is used for the analysis. In general, serum as well as plasma is acceptable for analysis. Nonetheless, higher or lower values may be found in specific types of material for ferritin, vitamin B12 and folate [44,51,53]. Therefore, if cut-offs are mentioned, the used sample type, analyser platform and method should be specified. Applying inappropriate cut-offs can lead to misinterpretation and may cause under- or overdiagnosis. Given the lack of harmonisation of ferritin, vitamin B12 and folate, at this point we recommend referring to local laboratory reference values or cut-offs provided by the local laboratory.

## Strengths and limitations

To our knowledge, this is the first study that systematically reviewed anaemia guidelines focusing on the diagnostics of the most common causes of anaemia. In addition to summarising diagnostic strategies, we present recommendations that may be used for developing new guidelines. A limitation of this study is that we only selected guidelines of countries with comparable healthcare systems that were available in Dutch, German or English, potentially missing other relevant guidelines. Nonetheless, our review provides an overview of diagnostic strategies from guidelines from all over the world.

## Conclusion

There was considerable variation in the included diagnostic criteria for anaemia caused by nutritional deficiencies in guidelines for the adult population. Even when similar diagnostic strategies were used, differences in cut-off values were seen and the supporting evidence was variable. Further research is needed to determine optimal algorithms. Even though laboratory tests are a fundamental aspect of the diagnostic algorithms, description of relevant laboratory aspects was lacking. For future guidelines, it will be important to include relevant laboratory aspects in guidelines, considering appropriate diagnostics and clinical decision limits.

## Supporting information

**S1 Table. Exclusion table.**
(XLSX)

**S2 Table. PRISMA abstract checklist.**
(DOCX)

**S3 Table. PRISMA checklist.**
(DOCX)

## Author contributions

**Conceptualization:** Margaretha G. M. Roemer, Wendy P. J. den Elzen, Bauke A. de Boer, Anne Margreet de Jong.

**Data curation:** Margaretha G. M. Roemer, Wendy P. J. den Elzen, Anne Margreet de Jong.

**Formal analysis:** Margaretha G. M. Roemer, Wendy P. J. den Elzen, Anne Margreet de Jong.

**Investigation:** Margaretha G. M. Roemer, Wendy P. J. den Elzen, Anne Margreet de Jong.

**Methodology:** Margaretha G. M. Roemer, Wendy P. J. den Elzen, Anne Margreet de Jong.

**Supervision:** Wendy P. J. den Elzen, Anne Margreet de Jong.

**Writing – original draft:** Margaretha G. M. Roemer, Wendy P. J. den Elzen, Anne Margreet de Jong.

**Writing – review & editing:** Margaretha G. M. Roemer, Wendy P. J. den Elzen, Rosalinde K. E. Poortvliet, Jacobijn Gussekloo, Steef Kurstjens, Bauke A. de Boer, Anne Margreet de Jong.

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
