## [Decision Letter · Decision Letter 0]

27 May 2025

PONE-D-25-00163Diagnostic algorithms in guidelines on anaemia: a systematic reviewPLOS ONE

Dear Dr. De Jong,

Thank you for submitting your manuscript to PLOS ONE. After careful consideration, we feel that it has merit but does not fully meet PLOS ONE’s publication criteria as it currently stands. Therefore, we invite you to submit a revised version of the manuscript that addresses the points raised during the review process.

**The reviewers suggested some minor amendments including some grammatical changes and in addition a few references to support the ferritin analysis. I also feel that the proposed title change  (review of nutritional anaemias) is a good suggestion.**

We look forward to receiving your revised manuscript.

Kind regards,

Elizabeth S. Mayne, M.D.

Academic Editor

PLOS ONE

Journal Requirements:

3. Please include a copy of Table 1, 2 & 3 which you refer to in your text on page 7.

4. Please include captions for your Supporting Information files at the end of your manuscript, and update any in-text citations to match accordingly. Please see our Supporting Information guidelines for more information: http://journals.plos.org/plosone/s/supporting-information .

5. As required by our policy on Data Availability, please ensure your manuscript or supplementary information includes the following:

Reviewers' comments:

Reviewer's Responses to Questions

**Comments to the Author**

1. Is the manuscript technically sound, and do the data support the conclusions?

Reviewer #1: Partly

Reviewer #2: Yes

2. Has the statistical analysis been performed appropriately and rigorously? 

Reviewer #1: N/A

Reviewer #2: Yes

3. Have the authors made all data underlying the findings in their manuscript fully available?

Reviewer #1: Yes

Reviewer #2: Yes

4. Is the manuscript presented in an intelligible fashion and written in standard English?

Reviewer #1: Yes

Reviewer #2: Yes

5. Review Comments to the Author

Reviewer #1: Thank you for the opportunity to review your manuscript that investigates similarities and differences in diagnostic algorithms of clinical treatment guidelines of nutritional anaemia in adult patients.

Major observations:

1. While many of the PRISMA 2020 checklist items were followed, I consider this manuscript to represent a narrative and thematic synthesis describing and comparing the guidelines and their recommendations. The quality of evidence and strength of recommendations contained in each clinical practice guideline were not evaluated using a recognised approach e.g., a rapid (iCAHE) or a complex instrument (the GRADE, AGREE II and AGREE-REX). In this respect I consider the methodology to fall short of a complete systematic review. I question the application of the scoring system used in the manuscript to rate the quality of the guidelines (p.5 3rd paragraph, 2nd sentence) as a suitable alternative. Table 1 shows three CPGs (8[2022], 14[2019], and 20[2018]) with similar diagnostic algorithms for iron deficiency yet variable evidence.

Minor observations:

1. I believe the title of the manuscript should be amended to reflect the focus of the work: e.g., Diagnostic algorithms in guidelines on nutritional anaemias in adults.

2. While Qatar’s health care system is a leader in the Gulf Cooperation Council, is it really similar to the healthcare systems of e.g., Western European countries in this study? Developing vs. developed health care system? As an example, in 2020 Qatar’s health care spending has exceeded its European counterparts but it was predominantly toward secondary care (70-75%), with ~20-25% for primary care and <5% for self-care. The comparative breakdown for the Netherlands was ~45%, ~45%, and ~10% for secondary, primary, and self-care, respectively.

3. p.4. 2nd paragraph 2nd last sentence: amend under- of overtreatment to under- or overtreatment

4. p.5 3rd paragraph 1st and 3rd sentence: repetitive: To ensure effective and standardised care, anaemia guidelines have been developed. Adhering to established clinical guidelines is advised to ensure effective and standardised care. Suggest consolidation.

5. p.5 3rd paragraph 4th sentence: I am unsure how physiological needs vary by ethnicity. Reconsider or substantiate.

6. References should be extensively reviewed to comply with the journal’s author guideline. In particular; 1. Use the Vancouver referencing style and, for consistency, restrict authors to the first six followed by et al. and 2. Review referencing style of internet resources. Some of the listed URLs are stale.

7. Abbreviations used in the tables should be explained in the caption e.g., <n; a=" " etc.=" " n=" ">8. The ferritin cut-off of reference 9 in Table 2 appears erroneous.</n;>

Reviewer #2: This is a highly relevant and useful article which is well-written.

Some minor grammatical edits are required, and additional suggested references are provided to include important pre-analytical factors affecting serum ferritin levels which are not widely recognized and can lead to diagnostic confusion.

6. PLOS authors have the option to publish the peer review history of their article (what does this mean? ). If published, this will include your full peer review and any attached files.

**Do you want your identity to be public for this peer review?** For information about this choice, including consent withdrawal, please see our Privacy Policy .

Reviewer #1: No

Reviewer #2: No

---

## [Author Response · Author response to Decision Letter 1]

10 Jul 2025

Comments of academic editor

The reviewers suggested some minor amendments including some grammatical changes and in addition a few references to support the ferritin analysis. I also feel that the proposed title change (review of nutritional anaemias) is a good suggestion.

Thank you for the evaluation of our manuscript. We agree with the minor amendments and proposed title change and have adjusted the manuscript accordingly.

Review comments to the Author

Reviewer #1: Thank you for the opportunity to review your manuscript that investigates similarities and differences in diagnostic algorithms of clinical treatment guidelines of nutritional anaemia in adult patients.

Major observations:

1. While many of the PRISMA 2020 checklist items were followed, I consider this manuscript to represent a narrative and thematic synthesis describing and comparing the guidelines and their recommendations. The quality of evidence and strength of recommendations contained in each clinical practice guideline were not evaluated using a recognised approach e.g., a rapid (iCAHE) or a complex instrument (the GRADE, AGREE II and AGREE-REX). In this respect I consider the methodology to fall short of a complete systematic review. I question the application of the scoring system used in the manuscript to rate the quality of the guidelines (p.5 3rd paragraph, 2nd sentence) as a suitable alternative. Table 1 shows three CPGs (8[2022], 14[2019], and 20[2018]) with similar diagnostic algorithms for iron deficiency yet variable evidence.

We acknowledge the lack of a recognized grading approach for the different guidelines. However, it is important to note that we did not evaluate guidelines as a whole, but specifically focused on the laboratory parameters within the guidelines and the evidence that was used to support the different diagnostic algorithms. Consequently, the same guideline may have a different quality for the different nutritional anaemias, e.g. iron deficiency anaemia (evidence = 2) and vitamin B12/folate deficiency anaemia (evidence = 3) in the Dutch anaemia guideline. Similarly, even though a similar diagnostic algorithm is used in different guidelines, the evidence supporting this algorithm may differ. To clarify this, we changed the following sentence in the discussion on page 10, line 10: “Furthermore, a lot of variation was seen in the recommended diagnostic criteria, cut-off values and evidence underpinning the algorithms, even in the presence of similar diagnostic strategies.”

We do agree that the wording in our title: “systematic review” may be misleading. Therefore, we removed this from the title and changed it to: “Comparison of diagnostic algorithms in guidelines on nutritional anaemias in adults”.

Minor observations:

1. I believe the title of the manuscript should be amended to reflect the focus of the work: e.g.,

Diagnostic algorithms in guidelines on nutritional anaemias in adults.

Thank you, we agree with this suggestion and have adjusted the title to: “Comparison of diagnostic algorithms in guidelines on nutritional anaemias in adults”.

2. While Qatar’s health care system is a leader in the Gulf Cooperation Council, is it really similar to the healthcare systems of e.g., Western European countries in this study? Developing vs. developed health care system? As an example, in 2020 Qatar’s health care spending has exceeded its European counterparts but it was predominantly toward secondary care (70-75%), with ~20-25% for primary care and <5% for self-care. The comparative breakdown for the Netherlands was ~45%, ~45%, and ~10% for secondary, primary, and self-care, respectively.

There is certainly a difference between health care systems in different countries, but this is true even within Western European countries. For instance: in the Dutch health care system, the general practitioner acts as a gatekeeper for referral to medical specialists, while in the German health care system, patients have direct access to medical specialists. Qatar has consistently ranked among the top healthcare systems globally and we therefore believe that it is justified to include the Qatar anaemia guideline in this manuscript.

3. p.4. 2nd paragraph 2nd last sentence: amend under- of overtreatment to under- or overtreatment

Done.

4. p.5 3rd paragraph 1st and 3rd sentence: repetitive: To ensure effective and standardised care, anaemia guidelines have been developed. Adhering to established clinical guidelines is advised to ensure effective and standardised care. Suggest consolidation.

Thank you for pointing this out. We removed the second sentence.

5. p.5 3rd paragraph 4th sentence: I am unsure how physiological needs vary by ethnicity. Reconsider or substantiate.

Agree. We reworded the sentence to: “For different physiological settings, including age, sex and pregnancy status or for specific patient groups, different guidelines for diagnosing anaemia have been developed.”

6. References should be extensively reviewed to comply with the journal’s author guideline. In particular; 1. Use the Vancouver referencing style and, for consistency, restrict authors to the first six followed by et al. and 2. Review referencing style of internet resources. Some of the listed URLs are stale.

Done, see also point 6 of journal requirements.

7. Abbreviations used in the tables should be explained in the caption e.g., 8. The ferritin cut-off of reference 9 in Table 2 appears erroneous.

We added a caption explaining the abbreviations to the tables.

Reviewer #2: This is a highly relevant and useful article which is well-written.

Some minor grammatical edits are required, and additional suggested references are provided to include important pre-analytical factors affecting serum ferritin levels which are not widely recognized and can lead to diagnostic confusion.

We would like to thank the reviewer for her/his appreciative comments and for the effort invested in evaluating our work. We incorporated the suggested edits in the attached document. See specifics below:

Edits in introduction:

• grammar: to find the underlying cause

• grammar: from ferritin only, to ferritin in various combinations...

• suggest reword: affecting up to 25%...

• add: red blood cells for the age and sex of the individual.

• reword: The diagnosis of anaemia typically includes ...

• suggest reword: physiological settings including age, sex, ... different guidelines for the diagnosis of anaemia have been developed.

Edits in discussion:

• the CBC

• suggest delete.

reword: we recommend including at least age and sex in the diagnostic algorithms

• when the same sample is measured

Edits in Table 1:

• The (capitalise)

• This international statement Hb specifically refers to the pre-operative management of anaemia. Please add this into the legend:

* International consensus statement for pre-operative anaemia: Hb< 13 g/dl for both sexes.

Edits in Figure 2:

• use MCV as a supportive parameter only

Furthermore, we incorporated the suggested references in the text.

Specifically, on page 10, we added the reference to the ferritin cut-off study to the following sentence on line 30: “For iron deficiency anaemia, ferritin was included in all algorithms. However, the used cut-offs were variable, as has been previously described”

In addition, we added the following sentence to the discussion, page 10, lines 35-38:

“Furthermore, ferritin levels are affected by recent oral iron supplementation intake, coffee and other dietary factors. Therefore, it is recommended to fast and withhold supplements before measuring ferritin. Nonetheless, these factors are not emphasized in guidelines, and therefore this can lead to diagnostic confusion.”

Journal requirements

Done.

Please confirm at this time whether or not your submission contains all raw data required to

replicate the results of your study.

Submission contains all data required.

There are no ethical or legal restrictions.

3. Please include a copy of Table 1, 2 & 3 which you refer to in your text on page 7.

Tables are placed in the manuscript file directly after the paragraph in which it is first cited, as described in the journal requirements.

Supporting Information files are updated.

Supporting information

Supplementary Table 1. Exclusion table

Supplementary Table 2. PRISMA abstract checklist

Supplementary Table 3. PRISMA checklist

5. As required by our policy on Data Availability, please ensure your manuscript or supplementary information includes the following:

A numbered table of all studies identified in the literature search, including those that were excluded from the analyses. For every excluded study, the table should list the reason(s) for exclusion.

We generated a table containing all identified guidelines in our database search. The dates of the search and date of assessment and consensus was added. Furthermore, we added the reason for exclusion. See Supplementary Table 1.

Not applicable

Not applicable

Done, reference list was updated.

---

## [Decision Letter · Decision Letter 1]

15 Aug 2025

PONE-D-25-00163R1Comparison of diagnostic algorithms in guidelines on nutritional anaemias in adultsPLOS ONE

Dear Dr. De Jong,

Thank you for submitting your manuscript to PLOS ONE. After careful consideration, we feel that it has merit but does not fully meet PLOS ONE’s publication criteria as it currently stands. Therefore, we invite you to submit a revised version of the manuscript that addresses the points raised during the review process.

There are some minor grammatical and typographical changes which need to be done and one of the reviewers has requested a review of the one table.

We look forward to receiving your revised manuscript.

Kind regards,

Elizabeth S. Mayne, M.D.

Academic Editor

PLOS ONE

Journal Requirements:

Reviewers' comments:

Reviewer's Responses to Questions

**Comments to the Author**

1. If the authors have adequately addressed your comments raised in a previous round of review and you feel that this manuscript is now acceptable for publication, you may indicate that here to bypass the “Comments to the Author” section, enter your conflict of interest statement in the “Confidential to Editor” section, and submit your "Accept" recommendation.

Reviewer #1: All comments have been addressed

Reviewer #2: (No Response)

2. Is the manuscript technically sound, and do the data support the conclusions?

Reviewer #1: Yes

Reviewer #2: Yes

3. Has the statistical analysis been performed appropriately and rigorously? 

Reviewer #1: N/A

Reviewer #2: Yes

4. Have the authors made all data underlying the findings in their manuscript fully available?

Reviewer #1: Yes

Reviewer #2: Yes

5. Is the manuscript presented in an intelligible fashion and written in standard English?

Reviewer #1: Yes

Reviewer #2: Yes

6. Review Comments to the Author

Reviewer #1: All my comments have been adequately addressed. I have no more concerns on the content of the manuscript.

Reviewer #2: I have further reviewed and provided detailed comments and edits on the manuscript in track changes.

Whilst improved, the manuscript still requires further minor revisions and another round of review.

7. PLOS authors have the option to publish the peer review history of their article (what does this mean? ). If published, this will include your full peer review and any attached files.

**Do you want your identity to be public for this peer review?** For information about this choice, including consent withdrawal, please see our Privacy Policy .

Reviewer #1: No

Reviewer #2: No

---

## [Author Response · Author response to Decision Letter 2]

29 Aug 2025

Reviewer #1: All my comments have been adequately addressed. I have no more concerns on the content of the manuscript.

Thank you for the evaluation of our revised manuscript.

Reviewer #2: I have further reviewed and provided detailed comments and edits on the manuscript in track changes.

Whilst improved, the manuscript still requires further minor revisions and another round of review.

Thank you for the evaluation of our revised manuscript.

We agree with most of the suggestions, which have been adjusted (for details see the manuscript with track changes). In brief, we changed the title, Figure 2 and most of the grammatical suggestions were incorporated. In addition we removed the term ‘systematic review’ in the whole manuscript, and clarified the terminology on reticulocyte counts and CRP/ESR. In addition the Footnotes for Tabel 2 and 3 are adjusted.

---

## [Decision Letter · Decision Letter 2]

18 Sep 2025

Comparison of diagnostic algorithms used in guidelines on nutritional anaemias in adults

PONE-D-25-00163R2

Dear Dr. De Jong,

We’re pleased to inform you that your manuscript has been judged scientifically suitable for publication and will be formally accepted for publication once it meets all outstanding technical requirements.

Kind regards,

Elizabeth S. Mayne, M.D.

Academic Editor

PLOS ONE

Additional Editor Comments (optional):

Reviewer #2:

Reviewers' comments:

Reviewer's Responses to Questions

**Comments to the Author**

1. If the authors have adequately addressed your comments raised in a previous round of review and you feel that this manuscript is now acceptable for publication, you may indicate that here to bypass the “Comments to the Author” section, enter your conflict of interest statement in the “Confidential to Editor” section, and submit your "Accept" recommendation.

Reviewer #2: (No Response)

2. Is the manuscript technically sound, and do the data support the conclusions?

Reviewer #2: Yes

3. Has the statistical analysis been performed appropriately and rigorously? 

Reviewer #2: N/A

4. Have the authors made all data underlying the findings in their manuscript fully available?

Reviewer #2: Yes

5. Is the manuscript presented in an intelligible fashion and written in standard English?

Reviewer #2: Yes

6. Review Comments to the Author

Reviewer #2: The corrections and comments have been addressed, and the article is now suitable for publication.

7. PLOS authors have the option to publish the peer review history of their article (what does this mean? ). If published, this will include your full peer review and any attached files.

**Do you want your identity to be public for this peer review?** For information about this choice, including consent withdrawal, please see our Privacy Policy .

Reviewer #2: No

---

## [Editor Report · Acceptance letter]

PONE-D-25-00163R2

PLOS ONE

Dear Dr. De Jong,

I'm pleased to inform you that your manuscript has been deemed suitable for publication in PLOS ONE. Congratulations! Your manuscript is now being handed over to our production team.

Kind regards,

on behalf of

Dr. Elizabeth S. Mayne

Academic Editor

PLOS ONE